# Cortisol Levels in Infants with Central Coordination Disorders during Vojta Therapy

**DOI:** 10.3390/children8121113

**Published:** 2021-12-02

**Authors:** Wojciech Kiebzak, Arkadiusz Żurawski, Stanisław Głuszek, Michał Kosztołowicz, Wioletta Adamus Białek

**Affiliations:** 1Institute of Health Science, Collegium Medicum, Jan Kochanowski University, 25-369 Kielce, Poland; wojciech.kiebzak@ujk.edu.pl; 2Laboratory of Medical Genetics, Department of Surgical Medicine, Collegium Medicum, Jan Kochanowski University, 25-317 Kielce, Poland; stanislaw.gluszek@ujk.edu.pl (S.G.); wioletta.adamus-bialek@ujk.edu.pl (W.A.B.); 3Sandomierskie Towarzystwo Naukowe, 27-600 Sandomierz, Poland; mkosztolowicz@onet.eu; 4Kieleckie Towarzystwo Naukowe, 25-303 Kielce, Poland

**Keywords:** cortisol, CCD, Vojta, physiotherapy

## Abstract

Background: Due to the decrease in the percentage of perinatal mortality, which is one of the Millennium Development Goals, the number of children with a central coordination disorder (CCD) has increased, present in up to 40% of premature babies. Neurodevelopmental disorders detected in the diagnostic process require early interventions that will eliminate or overcome existing dysfunctions. These treatments often cause discomfort in the infant, which induces insecurity and activation of basic defense mechanisms. The aim of the work is to assess changes in cortisol concentration in infants treated with the Vojta method. Methods and findings: The study included 35 children with CCD aged between three and nine months. The participants had no comorbidities that could have affected the obtained results. The activities were planned to occur in three stages: 1. Collection of a saliva sample directly before the physiotherapy appointment. 2. Collection of saliva immediately after rehabilitation. 3. Collection of saliva 20 min after the end of rehabilitation. The physiotherapeutic intervention included the assessment of seven reactions of the body position in space according to Vojta and the conduct of a therapeutic session consisting of the first phase of rotation and creeping reflex according to Vojta. The concentration of free cortisol in saliva was assessed with LC-MS/MS. In the first measurement, none of the children presented an excess of the normative concentration of cortisol. The cortisol measurement performed directly after rehabilitation showed above-normative values in three children. In the third measurement, all of the children presented a decreased concentration of free cortisol. The analysis (paired two-tailed *t*-test, *p* < 0.05) showed statistically significant differences between particular stages of the measurements. The analysis of the scores obtained in the second measurement showed the concentration of scores in the area of “normal” at a level of 0.83 (normal concentration) and the area “above normal” at the level of 0.005 (very weak concentration). Based on the analysis of significance of the obtained scores, it was found that the result was not accidental, and the Vojta method used in the treatment of children with CCD was suitable. Conclusions: Here, for the first time, we presented how Vojta therapy was correlated with cortisol levels among children with a central coordination disorder.

## 1. Introduction

Neurodevelopmental disorders are a serious problem affecting 3–4% of children [1]. It is estimated that in low-income countries such disorders develop in 0.5% of children, which results in high infant mortality [2]. Improving the survival of premature babies, which is one of the Millennium Development Goals [3], is a challenge in comprehensive care for a developing child. The number of children with a central coordination disorder (CCD) in this group varies between 23 and 42%, and the smaller the number, the more mature the born baby [4].

Diagnosing developmental disorders at an early stage of life is difficult [5]. The planned process of CCD detection allowing to assess dynamics and direction of changes in the aforementioned disorders of children’s development is a special part of such activities [6]. Many tools are used in the diagnostic process, including Prechtl, Hellbrüge, or Vojta diagnostics. Neurodevelopmental disorders detected in the diagnostic process require early interventions to eliminate or overcome the dysfunctions [7,8,9,10]. Improving social contacts and the quality of neurological reflexes that precede the improvement of spontaneous motor skills and changes in postural responses are the first effects of an early Vojta intervention [11]. In the authors’ view, observation of children with CCD should be long-term because it allows for detecting significant dependencies, e.g., those concerning visual perception [12]. It should be emphasized that the main goal of diagnostic and therapeutic activities, apart from obtaining proper clinical parameters, is to improve the quality of life.

According to the authors, the aforementioned treatment is often associated with keeping a forced position or stretching muscles in rehabilitated children [13]. These treatments often cause discomfort to the child, which in turn induces insecurity and activation of the body’s basic defense mechanisms, primarily the secretion of glucocorticosteroids (stress hormones), then adrenaline and norepinephrine, which in turn enhance the action of cortisol. Cortisol regulates many physiological and pathophysiological processes, including the response to stress by generating energy, e.g., through the release of glucose into the blood, stimulating glucogenesis, increasing blood pressure, and changing mood and behavior to defend against emerging danger [14]. The expression of cortisol is constitutive but variable, characterized by a diurnal and pulsatile rhythm [15], with the highest concentration observed between 10 a.m. and 12 a.m. and the lowest between 2 a.m. and 4 a.m. at night. [16]. Cortisol is produced by the zona fasciculata of the adrenal cortex, but its synthesis is regulated by the adrenocorticotropic hormone (ACTH) secreted by the pituitary gland. A chronic excess cortisol level in the blood is the main reason for the development of Cushing’s syndrome [17] but also for lowering natural resistance to infections, delayed wound healing, obesity, or damage to hippocampus cells, the latter of which, in turn, disturbs the development of cognitive processes [18,19,20,21]. On the other hand, attention should also be drawn to too low cortisol levels, which may reflect underdevelopment of the adrenal cortex and lead to overexpression of other hormones. In such a situation, constant stimulation of the hypothalamic–pituitary system to increase the secretion of corticoliberin (CRH) and corticotropin (ACTH) is observed, with CRH and ACTH stimulating the adrenal cortex and causing its hypertrophy. Therefore, routine cortisol testing in children with nervous system disorders should be considered. Especially in this case, it is important to analyze whether applied physiotherapy procedures in such children with CCD do not induce disorders in secretion and/or maintenance of normal cortisol levels in the body, which could have further negative consequences.

## 2. Materials and Methods

### 2.1. Study Project

The study assumes the assessment of the concentration of free cortisol in the saliva of a-few-month-old infants subjected to Vojta stimulation in the preplanned stages: 1. Collection of a saliva sample directly before an appointment with a physiotherapist. 2. Collection of saliva directly after rehabilitation. 3. Collection of saliva 20 min after a physiotherapeutic intervention has finished. All the activities were performed in the morning.

### 2.2. Ethical Approval

The study was approved by the Bioethics Committee (Approval No. 28/2020 CM UJK). The children’s legal guardians gave their written approval to participate in the study.

### 2.3. Physiotherapeutic Intervention

The physiotherapeutic intervention included the assessment of seven reactions of the body position in space by Vojta and a therapeutic session consisting of the first phase of rotation and creeping reflex according to Vojta [13]. Stimulation time in each position was 45 s and was repeated twice on each side; total stimulation time was about 6 min. The stimulation was performed by a certified Vojta physiotherapist. The total intervention time was approximately 20 min [13].

### 2.4. Subjects

Thirty-five infants with appropriately selected inclusion and exclusion criteria qualified for the study (Table 1).

On the day of the study, the children were in the age range of 3–9 months (mean 5.46 ± 1.82). Birth weight of the subjects ranged between 2570 and 4850 g (mean 3260 ± 560), 14 of whom were delivered by C-section. All examined children were born at term (38–40 weeks of gestation). All the children were born without complications. Two scored 8 Apgar points at birth, nine gained 9 points, and the remaining—10 Apgar points. The medical diagnoses in the referral letter concerned ICD 10: R29.8 (other and unspecified symptoms and signs involving the nervous and musculoskeletal systems) and R62 (lack of expected normal physiological development).

The children qualified for the study had no comorbidities and were healthy on the day of the study. Health condition assessment necessary for the qualification to implement rehabilitation treatment was performed by a pediatrician. To ensure the homogeneity of the group, the study included children diagnosed with moderate-severe CCD according to Vojta [13]. The children were only diagnosed with CCD; none of the examined children showed features of developmental delay. The researchers had no influence on the order of children’s admission; selection of the study group was random.

### 2.5. The Analysis of Cortisol Concentration in Saliva

The concentration of free cortisol in saliva was assessed with the LC-MS/MS method [22]. A sterile Salivette collection kit (Sarstedt AG & Co., Numbrecht, Germany) containing a sponge was placed in the infant’s mouth and used to collect saliva. The child chewing a sponge soaked it with saliva, allowing the collection of the amount of biological material necessary for the measurement. Material for the tests was collected under sterile conditions by a trained person not involved in therapeutic activities. After collection, the material was preserved and transported to the laboratory, assessing the free cortisol concentration in saliva.

The norms for the minimum and maximum cortisol levels in children in the studied age were determined based on literature data [23,24].

### 2.6. Statistical Analysis

To analyze the obtained results, basic descriptive statistics were calculated: the mean, standard deviation, variance, and minimum and maximum values.

Friedman’s ANOVA and Kendall’s coefficient of concordance was used, which allowed for the simultaneous comparison of several consecutive measurements.

For the assessment of measurement scores, it is necessary to determine the concentration of individuals in elementary areas: “normal” (*n*_1_) and “above normal” (*n*_2_), by calculating the ratio of the population size (*n*) for the appropriate elementary field to the combination of the sum of the size. For this purpose, the formula was used: C(*n*_1_) = [(*n*_1_ − 1) × *n*_1_]/[(*n*_1_ + *n*_2_ − 1) × (*n*_1_ + *n*_2_)]. To test the significance of the obtained scores, an analog of the structural index was used [12]:U = (C*n*_1_ − C*n*_2_)/√P × Q/*n*, where *p* = (C*n*_1_ + C*n*_2_) × (1 − (C*n*_1_ + C*n*_2_))

## 3. Results

In the first measurement, the majority of children (20/35) showed a cortisol level below the minimum value (<3.5 nmol/L), and the remaining children, except four whose level was within the normal range, showed a borderline level; also, none of the children showed a score above the normative cortisol concentration [23,24] (Figure 1).

There was no statistically significant correlation between the child’s birth weight, age, and cortisol level secreted. The cortisol measurement performed directly after the intervention exceeded the normative values in three children (>40 nmol/L). In the third measurement, all of the children presented normal concentrations of free cortisol. It was observed that despite the low level of cortisol in individual children, its production was efficiently stimulated. The results’ analysis via Friedman’s ANOVA indicated statistically significant differences (*p* < 0.0001) between the individual stages of the measurements for each sample individually (Figure 2).

The cortisol concentration in children after the intervention increased from 1.5–1.6 (2 patients) to 18.8 (1 patient) times compared to that in the first measurement; the mean cortisol level was 12.5 nmol/L. After 20 min, a significant decrease in the concentration of the hormone was observed in relation to the second measurement with the average value of 9.5 nmol/l. The decrease in concentration in the third measurement compared to the second measurement ranged from 1.01 to 3.4 (Figure 3).

Considering the analysis of the variability of cortisol secretion in individual children (Figure 1), it was shown that in 31% of the subjects, the increase in cortisol concentration in saliva after stimulation was reasonable; in 10% of the infants, a significantly higher increase in the level of cortisol was observed compared to that of the rest. The fluctuating cortisol level was clearly visible at all three stages of the study in all the children tested.

The analysis of the concentration of scores in the first and third measurements showed the accumulation of scores only in the area “normal”, while in the second measurement, there was also an accumulation of results in the area “above the norm”. The analysis of the scores obtained in the second measurement showed the concentration of scores in the area “normal” at the level of 0.83, which was a normal concentration, and in the area “above normal” at the level of 0.005, which was a very low concentration. On the basis of the significance test, the obtained score was U = 5.89 > 1.96. H0 was rejected, which means that the obtained positive score was not accidental, and the method used in the treatment of children with CCD was appropriate and probably did not result in exceeding the normative level of cortisol.

## 4. Discussion

In the process of early diagnosis and therapy in high-risk infants with a brain injury, the choice of intervention methods is difficult [25]. The vast majority of published studies on the treatment of high-risk children with CCD emphasize the main effects and group differences while paying attention to individual differences. It should be recognized that the results at the group level may not be fully applicable to each member of this group, especially in the face of increasingly discernible differences at the genome level [26]. It was shown that certain genetic factors or genetic polymorphisms may be associated with the risk of CCD [27]. CCD is a multifactorial and clinically heterogeneous group of disorders; still, a few research studies indicate how particular genetic profiles may influence the response to motor interventions [28,29]. A broader approach to treating children with CCD is worth considering, e.g., by looking into genetic factors or epigenetic factors such as stress or basic childhood personality types, and how they interact with recovery or adaptation to brain damage, or creatively looking for other potential factors that have not yet been identified [30]. The ultimate goal of diagnostic and treatment activities is to transform the intervention prescription. A universal approach needs to be changed to an evidence-based individual care plan in which each child and family may choose to participate or receive only those interventions that can maximize therapeutic benefit. The mentioned prescription should be consistent with their personal life goals and desires [31]. Therapeutic activities often cause discomfort and are a source of stress [32], which may either stimulate or impair development, depending on the individual characteristics of the patient, especially with a patient with neurological disorders.

The presented results of our study showed, for the first time, a clear influence of physiotherapeutic intervention on the change in cortisol levels in children (Figure 2). However, the changes in the level of the described hormone were different in particular children (Figure 1). Literature data indicate that in most healthy infants, cortisol levels range from 4.4 to 25 nmol/L [23], although a significant dependence, e.g., on the time of the day, should also be considered. In the conducted study, to interpret the scores, the minimum cortisol level was set at 3.5 nmol/L. This assumption was made due to lack of information on the range of the norm for the studied children and relatively low birth weight in some children (<3000 g); the maximum level was set at 27.8 nmol/L, the value given by Tollenar et al. [23] for five-month-old children. This assumption made it possible to observe that most of the children had too-low cortisol levels, which prompts further research on the problem. Can CCD be correlated with a lowered level of basal cortisol among children, and can it reflect an inadequate stress response? It is worth considering the cause of our observations: is it related to the mechanism of endocrine glands’ activity? The reduced level of baseline cortisol resulted in its lower production during the stress condition compared to that in children whose baseline cortisol level was appropriate. On the other hand, multiple increases in cortisol levels were observed in a stressful situation, which contrasts with that in healthy people (usually, there is a two-fold increase in cortisol) described in the literature [33]. Long-term follow-up studies could answer whether the abnormal dynamics of cortisol level is typical or incidental and whether it will have further health consequences for developing children [34,35]. However, it should be emphasized that in the study, despite this low cortisol concentration, its level increased significantly after rehabilitation, and 20 min after it was finished, it remained at the normative level. It was not determined how long it would remain normal, but the applied method seemed beneficial from the point of view of stimulating the organism for the adrenal cortex to function properly. The lack of clear data on the dynamics of cortisol in healthy children subjected to severe stress, and all the more so in children with neurological disorders, hinders their proper clinical assessment. The literature reports that in children aged from four to nine months, the stress response causes a 50% average increase in the cortisol level [33]. In the case of our study, 94% of children showed more than a two-fold increase in cortisol concentration in response to stress; on average, a 4.6-fold increase in cortisol concentration with a very high standard deviation was observed. Such a result confirms the assumption made by Buford et al. that the scores at the group level may not be fully applicable to each member of this group [26], indicating that therapeutic decisions should be more individualized.

Cortisol secretion is a mechanism necessary for the proper development of a child and their functioning, especially in a situation potentially disrupting the sense of safety. Most commonly known as the “stress hormone”, cortisol affects many functions in people. It is involved in regulating blood pressure; the immune system; the metabolism of proteins, carbohydrates, and fats; and has an anti-inflammatory effect. Since the proper balance of cortisol is essential to human health, it is vital to monitor the level of cortisol in the human body. Cortisol levels are usually monitored in blood, plasma, serum, saliva, sweat, and hair samples by employing immunochemical and analytical methods [36]. Here, we present, for the first time, evidence that a physiotherapeutic intervention resulted in a significant temporary increase in cortisol level; however, a study conducted by Sezer et al. [37] revealed that regular use of rehabilitation significantly reduces the level of this hormone in the body on a daily basis. The cited authors demonstrated a positive effect of daily physiotherapeutic activities on improving the bone density and cortisol concentrations in children hospitalized in the neonatal ward, in whom the intervention was performed for 30 consecutive days. The continuation of our study on the same children could be a valuable reference point to the results presented by Sezer et al. [37].

The authors of the study noticed the stress factor accompanying the response to physiotherapy in infants, manifested by an increased cortisol level, which was confirmed by the scores obtained in the study. Cortisol is one of the elements of the integrated action mechanisms of the hypothalamus–pituitary–adrenal axis, which determines many events in the human body that are seemingly independent of each other. Therefore, we believe that the observed phenomenon should be carefully examined because the dysfunction of one element can affect the others [38,39,40]. In addition, it would need to be verified whether a non-normative cortisol level and/or its dynamics will correlate with abnormalities in the production of other hormones regulated by the same mechanism, influencing, e.g., dysfunctions in fluid and electrolyte balance. It is believed that the magnitude of cortisol response to stress is regulated by the interaction of environmental and genetic determinants [41]. The obtained scores illustrated how different the response was among particular children to the same form of stimulation (Figure 1), which prompts further study, especially in the direction of genetic determinants acquired during fetal life or other factors that could have influenced a different response to stress in the studied children. The sources of such significant differences may be diverse, and, importantly, several studies indicate that the quality of early maternal care impacts individual responses of cortisol and dopamine to stress throughout life [42,43].

In conclusion, it should be emphasized that the therapeutic intervention was associated with a significant increase in the concentration of cortisol in saliva; however, the increase was brief and within the normal concentration range. Heterogeneous results indicate the need to take this parameter into account in the process of planning therapeutic activities. Further research is required to assess the response to other forms of physiotherapeutic stimulation, and an extension of the test period will show the kinetics of cortisol secretion over a longer period of time (following up the observation of the same patients during rehabilitation).

### 4.1. Limitations

The presented results reflect a pilot study; hence, the sample size is limited and does not allow for inference about the entire population. Further study should be conducted on a bigger sample, with a control group, and assessing the concentration of other hormones involved in regulating the body’s physiological response to stress. A more complete assessment of the dynamics of cortisol secretion will be an important aspect, especially in children with low cortisol levels or, e.g., in children born prematurely and/or with low birth weight. In addition, it is advisable to evaluate children’s behavior during and after treatment (comfort) and correlate the result with cortisol outcomes.

### 4.2. Clinical Implications

The presented results show the effect of Vojta stimulation on a brief increase in cortisol levels in infants with CCD. The obtained results indicate the need to consider a stress factor in planning the therapy of children with CCD.

## 5. Conclusions

Vojta stimulation increased the level of free cortisol in saliva among infants with CCD, which, in 8.57% of children after therapy, exceeded normative values.After the Vojta intervention, the level of free cortisol in saliva decreased significantly, reaching reference values after 20 min.

## Figures and Tables

**Figure 1 children-08-01113-f001:**
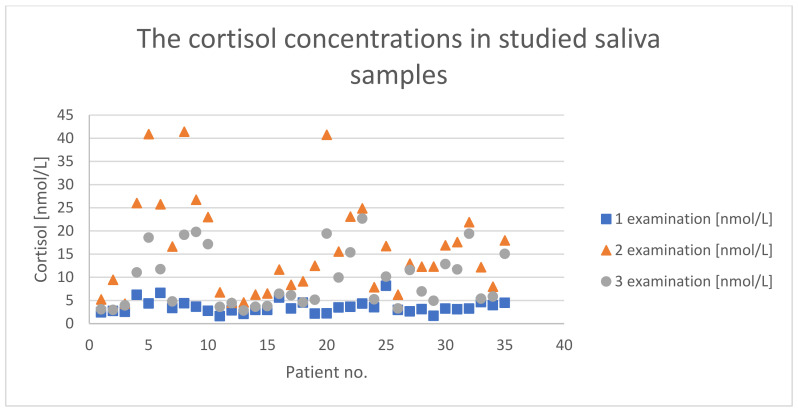
Cortisol concentrations in the studied saliva samples collected from 35 infants during the particular study stages (first—before; second—during; third—after the rehabilitation treatment).

**Figure 2 children-08-01113-f002:**
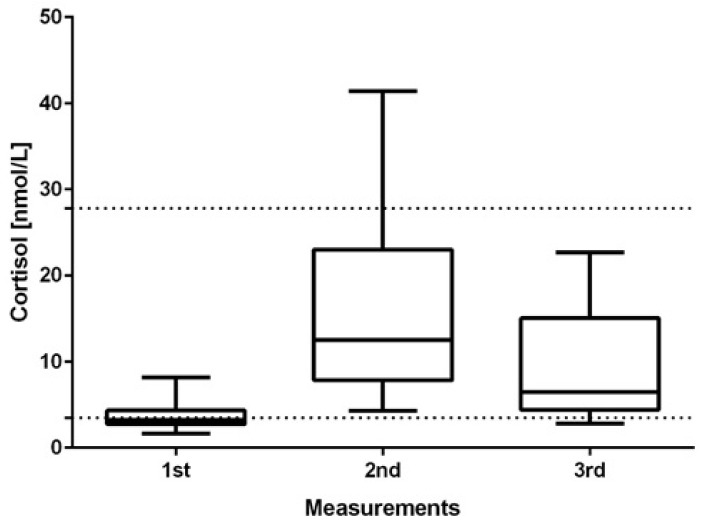
The comparison of cortisol concentrations in the saliva samples of the infants acquired before, during, and after the rehabilitation by Vojta’s method. The statistically significant differences (*p* < 0.05) were analyzed via a two-tailed paired *t*-test. The lines indicate the normal range between minimum and maximum values of the cortisol concentration.

**Figure 3 children-08-01113-f003:**
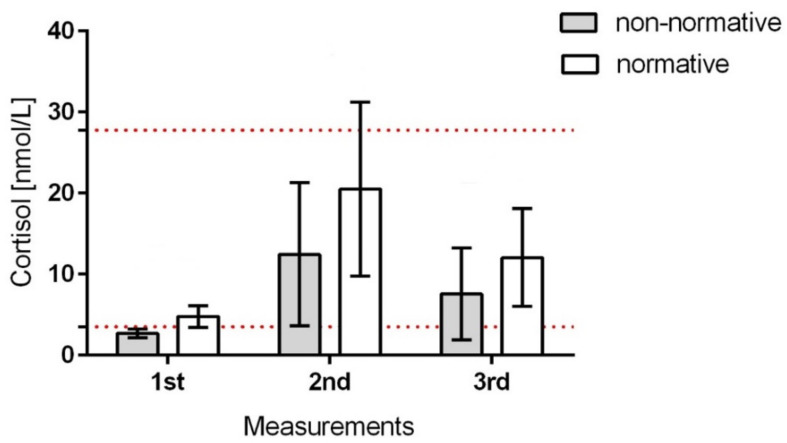
The mean cortisol concentrations in studied saliva samples collected from 20 infants who produced non-normative baseline cortisol in comparison to 15 infants who produced normative baseline cortisol (first measurements) and the next measurements (second—during treatment; third—after the rehabilitation treatment). The patients were marked with their birth weight and arranged on a scale from lowest to the highest; the dashed lines on the graph indicate the limits of the normative values of cortisol levels (minimum—3.5 nmol/L and maximum—27.8 nmol/L) for children aged from 3 to 9 months in the morning according to Ivars et al. (2015) and Tollenar et al. (2010).

**Table 1 children-08-01113-t001:** Inclusion and exclusion criteria in the study.

Inclusion Criteria	Exclusion Criteria
Aged 3–9 months Medium-severe CCD No comorbidities	No guardian’s consent to participate in the study
	Presence of disorders preventing the implementation of Vojta therapy [Vojta 2007]
	Up to 5 days from becoming vaccinated
	Feeding less than 30 min from the intervention

## Data Availability

All the data for the study are available from the authors without any restrictions.

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
