# Peer review of "Cortisol Levels in Infants with Central Coordination Disorders during Vojta Therapy"

_children, 2021, doi:10.3390/children8121113_

Round 1

Reviewer 1 Report

Much needid work both  measurement of cortisol  saliva in very young children and Vojta method.

Author Response

Dear Reviewer,
Thank you for your favorable opinion. The language has been verified and corrected again.
Regards,
Autors

Reviewer 2 Report

The authors examined the level of free cortisol in the saliva of 35 infants aged 3-9 months. The infants were diagnosed with moderate to severe central coordination disorder (CCD) according to the Vojta diagnostic scheme. The infants underwent Vojta physiotherapy treatment and cortisol was assessed immediately before the treatment session, at the end of the session and 20 minutes afterwards. The authors report low baseline levels of salivary cortisol (in the lower range of normal according to the literature), a variable increase after physiotherapy and a decrease to normal levels (although not to the low baseline level) after 20 minutes. The authors discuss these findings as a stress response to physiotherapy and place them in a broad context of infant development and endocrine system.

Although these results are helpful in understanding the stress response of infants to this particular type of physiotherapy, I see major weaknesses that would need to be addressed before the manuscript could be accepted for publication:

  1. the CCD is not a diagnosis but a construct of a series of non-optimal postural responses in a series of 7 tests according to Vojta; this construct was formulated before the advent of neuroimaging in infants to provide a prognosis. Today, however, it seems rather outdated. To understand the individually very different cortisol response of infants, a listing of the basic clinical diagnoses would be necessary (prematurity, failure to thrive, respiratory distress and artificial ventilation, stay in a neonatal intensive care unit, post-asphyxial encephalopathy and so on...).
  2. Do the infants only show CCD or are they also developmentally delayed? What does "exclusion of comorbidities" mean - also developmental comorbidities?
  3. As far as statistics are concerned, the cortisol data are clearly not normally distributed when looking at the boxplots. Therefore, the t-test is not appropriate. Furthermore, the t-test only examines 2 dependent values and not three as in this case. Obviously, multiple t-tests have been conducted. Therefore, a correction for multiple testing would have been necessary. I suggest using a non-parametric test that allows multiple dependent variables to be tested.
  4. I do not understand the method of calculating the scores, nor the need to do so. The stress response of cortisol is not abnormal, but a normal everyday occurrence, even in children.
  5. It is true and also my personal experience that most, but not all, children undergoing physiotherapy according to Vojta show stress symptoms (crying) to varying degrees. Thus, different stress reactions of the adrenal system would also be expected. In my opinion, it would have been necessary to assess the children's behaviour during and after treatment (comfort) and correlate it with the cortisol findings.
  6. The authors found the baseline values unexpectedly low compared to literature data, but all infants showed a stress response. I would not interpret the low baseline values as reflecting adrenal insufficiency in CCD as the authors discuss, but it appears to be an effect of normal values in their laboratory. I suggest examining baseline salivary cortisol in a series of healthy infants to obtain their own normal values. Adrenal insufficiency would only be expected in children with severe brain damage (including to the hypothalamus) or genetic syndromes with adrenal insufficiency - such children usually show recurrent hypoglycaemic episodes at this age, which does not appear to have been the case in the infants included in this study.
  7. The English language is weak and sometimes difficult to understand.
  8. As far as the endocrinological aspects are concerned, I think the introduction and discussion are overloaded. As I have pointed out above, the cortisol stress reaction found here is physiological and would occur in the same way in other stressful situations such as hunger, fever, vaccinations, crying for psychological reasons. It is therefore not expected that this cortisol response alone could affect neurodevelopment. This is not comparable to the constant noisy and stressful stimulation to which premature babies are exposed for weeks when they are cared for in the neonatal intensive care unit (NICU); here appropriate treatment and "gentle" physiotherapeutic measures have been shown to improve outcome. The risk of the Vojta method, in my opinion, is not the physiological stress response, but the disruption of the mother-child relationship that sometimes accompanies it.
  9. „Chronic insufficient cortisol level could overstimulate adrenal cortex and cause secondary growth of e.g. androgens and aldosterone. Such a condition may have serious health consequences in the developing child“. This applies to cases of adrogenital syndrome with genetically impaired cortisol synthesis, but not in other situations of adrenal insufficiency or insufficiency as in these children.
  10. In the introduction, it would be desirable to have more information on tests and diagnoses in the field of neurodevelopment with current literature.
  11. From an ethical and data protection point of view, it could be criticised that in Figure 1 the individual infants are identifiable by their birth weight; since birth weight apparently has no influence on the data, this should be replaced by a pseudonymous study number.
  12. If you want to show the ratios and not the differences in the cortisol values, I suggest to relate the changes for examination 2 and 3 respectively to the baseline values (Invest. 1) and not 3 to 2.
  13. The authors conclude at the end of their discussion: „The presented results explicitly show the effect of Vojta stimulation on a brief increase in cortisol levels in infants with CCD. The obtained results indicate the need to consider a stress factor in planning the therapy of children with CCD.“ This is correct and I support this interpretation - and as already indicated, the entire interpretation and discussion of the manuscript should be shortened and focused on this central finding and recommendation.

Author Response

Dear Reviewer,

Thank you for the critical assessment of the submitted text. We analyzed all the objections raised. We have made every effort to ensure that all doubts are clarified.

Below, let me refer to the individual remarks:

  1. the CCD is not a diagnosis but a construct of a series of non-optimal postural responses in a series of 7 tests according to Vojta; this construct was formulated before the advent of neuroimaging in infants to provide a prognosis. Today, however, it seems rather outdated. To understand the individually very different cortisol response of infants, a listing of the basic clinical diagnoses would be necessary (prematurity, failure to thrive, respiratory distress and artificial ventilation, stay in a neonatal intensive care unit, post-asphyxial encephalopathy and so on...).

Answer:

The 7 tests proposed by Vojta were used to ensure consistency between examination and treatment. We realize that much more precise tools are now available, such as neuromapping the brain. The very performance of the test, with rapid changes in the position of the child, can be stressful even for the parent standing next to it. In the Vojta method, these tests are an integral part of the appointment, so we did not want to separate these issues.

As suggested, the IDC 10 children's health and diagnosis information has been included in the text, for which we kindly thank you.

  1. Do the infants only show CCD or are they also developmentally delayed? What does "exclusion of comorbidities" mean - also developmental comorbidities?

Answer:

The children in the study were not developmentally delayed. They only showed symptoms of medium-severe CCD. Appropriate information has been included in the text. Thank you for your valuable comment, we apologize that this was not included in the original version.

  1. As far as statistics are concerned, the cortisol data are clearly not normally distributed when looking at the boxplots. Therefore, the t-test is not appropriate. Furthermore, the t-test only examines 2 dependent values and not three as in this case. Obviously, multiple t-tests have been conducted. Therefore, a correction for multiple testing would have been necessary. I suggest using a non-parametric test that allows multiple dependent variables to be tested.

Answer:

Thank you for this suggestion. The calculation method has been changed. The revised version uses the ANOVA Friedman statistic and the Kendal coefficient of concordance, which allow for the simultaneous comparison of several consecutive measurements.

  1. I do not understand the method of calculating the scores, nor the need to do so. The stress response of cortisol is not abnormal, but a normal everyday occurrence, even in children.

Answer:

We agree that stress is a natural phenomenon in children. It is also the result of many daily activities. Statistics were conducted to confirm that there were no random differences in cortisol levels. For the authors, it is particularly important that the level of cortisol decreases relatively quickly to the normative value after the therapy.

  1. It is true and also my personal experience that most, but not all, children undergoing physiotherapy according to Vojta show stress symptoms (crying) to varying degrees. Thus, different stress reactions of the adrenal system would also be expected. In my opinion, it would have been necessary to assess the children's behaviour during and after treatment (comfort) and correlate it with the cortisol findings.

Answer:

Thank you for your extremely valuable  observation. The results presented in the paper are a pilot study. Further analyzes are planned on a larger population, using a control group and taking into account more variables. The presented comment will certainly be taken into account in their planning. At this stage, this important limitation of research has been highlighted in the text as requiring further analysis.

  1. The authors found the baseline values unexpectedly low compared to literature data, but all infants showed a stress response. I would not interpret the low baseline values as reflecting adrenal insufficiency in CCD as the authors discuss, but it appears to be an effect of normal values in their laboratory. I suggest examining baseline salivary cortisol in a series of healthy infants to obtain their own normal values. Adrenal insufficiency would only be expected in children with severe brain damage (including to the hypothalamus) or genetic syndromes with adrenal insufficiency - such children usually show recurrent hypoglycaemic episodes at this age, which does not appear to have been the case in the infants included in this study.

Answer:

Thank you for your valuable  observation. As mentioned in the previous comment, studies including the analysis of results in the control group are already planned. This information has been included in the test limitation section.

  1. The English language is weak and sometimes difficult to understand.

Answer:

Thank you for this attention. The text was translated by a professional translator, but after introducing changes, it was re-analyzed by a specialist in this field.

  1. As far as the endocrinological aspects are concerned, I think the introduction and discussion are overloaded. As I have pointed out above, the cortisol stress reaction found here is physiological and would occur in the same way in other stressful situations such as hunger, fever, vaccinations, crying for psychological reasons. It is therefore not expected that this cortisol response alone could affect neurodevelopment. This is not comparable to the constant noisy and stressful stimulation to which premature babies are exposed for weeks when they are cared for in the neonatal intensive care unit (NICU); here appropriate treatment and "gentle" physiotherapeutic measures have been shown to improve outcome. The risk of the Vojta method, in my opinion, is not the physiological stress response, but the disruption of the mother-child relationship that sometimes accompanies it.

Answer:

We appreciate a very insightful look at our work. As we explained above – we refer to the standardized normative cortisol level and compare our results to this according to recommended methods. Additionally, to exclude some additional variables, we increased the normative ranges for cortisol. A high stress level generated during the Vojta method is present due to very "aggressive" treatment performed towards the child, involving the use of force and  performing often painful and above all sudden exercises, which the child does not experience on a daily basis, and in no case can it be compared with the absence of the mother. However, even if stress-induced cortisol levels are within the normal range, the observed baseline levels are below normal, which may have implications for a less efficient response to stress. What’s  more, the referenced standards for cortisol level  are related to normal conditions not stress, so we should suspect that the level of cortisol in healthy children during the under stress should be higher, especially that the cortisol level was measured during it’s the  its highest daily production (according to literature data). According to this, the patients observed in our study did not reflect the typical reactions. We mentioned that the lack of clear data on the dynamics of cortisol in healthy children subjected to severe stress, and more so in children with neurological disorders, hinders their proper clinical assessment, so we hope that our research will encourage the scientific community for to  further research.

  1. „Chronic insufficient cortisol level could overstimulate adrenal cortex and cause secondary growth of e.g. androgens and aldosterone. Such a condition may have serious health consequences in the developing child“. This applies to cases of adrogenital syndrome with genetically impaired cortisol synthesis, but not in other situations of adrenal insufficiency or insufficiency as in these children.

Answer:

Thank you very much for your comment, we agree that this situation relates to specific determinants, not exactly adequate to the presented issue. We tried to improve our statement. However, we observed 8/35 (22%) patients with insufficient or bordeline cortisol production, which in combination with CCD may reflect severe neurological disorders. Therefore, we emphasize the importance of monitoring cortisol levels in children with CCD. What’s more, we observed three cases of excessive production of cortisol in a stressful situation with the reduced concentration of baseline cortisol, which is also puzzling.

We would like to emphasize that for the first time we discovered that most CCD patients produce reduced baseline cortisol (57% of studied patients with CCD),which is the most important for the further studies and clinical observation of this group of patients.

  1. In the introduction, it would be desirable to have more information on tests and diagnoses in the field of neurodevelopment with current literature.

Answer:

Thank you for this comment. The introduction mentions three diagnostic tools available to the physiotherapist (Prehtl, Hellbruge, Vojta). These three tools have five references (numbers 7-11). The aim of the study was to show the process of changes in the level of cortisol in response to a therapeutic stimulus, therefore it was decided to limit some theoretical content.

  1. From an ethical and data protection point of view, it could be criticised that in Figure 1 the individual infants are identifiable by their birth weight; since birth weight apparently has no influence on the data, this should be replaced by a pseudonymous study number.

Answer:

Thank you for your comment, the chart has been changed as suggested.

  1. If you want to show the ratios and not the differences in the cortisol values, I suggest to relate the changes for examination 2 and 3 respectively to the baseline values (Invest. 1) and not 3 to 2.

Answer:

Thank you for your comment, the figure has been modified as recommended.

  1. The authors conclude at the end of their discussion: „The presented results explicitly show the effect of Vojta stimulation on a brief increase in cortisol levels in infants with CCD. The obtained results indicate the need to consider a stress factor in planning the therapy of children with CCD.“ This is correct and I support this interpretation - and as already indicated, the entire interpretation and discussion of the manuscript should be shortened and focused on this central finding and recommendation.

Answer:

Thank you for your observation. We hope that after the introduced changes, the content of the manuscript will be clearer and the conclusions drawn from the work will be better focused.

Thank you once again for all comments, we have made every effort to ensure that in the revised version all of them are satisfactorily addressed. We hope that the current version of the manuscript is better and meets the reviewer’s expectations.

Round 2

Reviewer 2 Report

The authors have answered all questions of this reviewer. They stress, that this is a pilot study (for example, the clinical stress-correlate to the cortisol level was not investigated, and the study is lacking a control group). Due to these deficiencies there is a high risk of bias in their results.

This risk of bias should be adressed more clearly in their discussion, and too far-going conclusions (such as negative consequences for development of the children due to the findíngs) should be avoided.

Author Response

Drogi recenzenciu,
DziÄ™kujÄ™ za dalsze wnikliwe komentarze. ZredagowaliÅ›my tekst, aby uniknąć zbyt daleko idÄ…cych wniosków.
Tekst do wydania jÄ™zykowego wysÅ‚aliÅ›my również do firmy zaproponowanej przez wydawnictwo. Angielski powinien być teraz Å‚atwiejszy do odczytania.
Mamy nadzieję, że tekst w obecnej formie spełni oczekiwania wobec publikacji w Children.

Pozdrawiam

Autorzy